# Impact of urban heat island on inorganic aerosol in the lower free troposphere: a case study in Hangzhou, China

Hanqing Kang[1,2,3], Bin Zhu[1,2], Gerrit de Leeuw[1,4,5,6], Bu Yu[7], Ronald J. van der A[1,4], Wen Lu[1,2]

[1]Key Laboratory for Aerosol-Cloud-Precipitation of China Meteorological Administration, Nanjing University of Information Science and Technology, Nanjing, 210044, China

[2]Collaborative Innovation Center on Forecast and Evaluation of Meteorological Disaster, Nanjing University of Information Science and Technology, Nanjing, 210044, China

[3]Chongqing Institute of Meteorological Sciences, Chongqing, 401147, China

[4]KNMI (Royal Netherlands Meteorological Institute), R&D Satellite Observations, P.O.Box 201, 3730AE  De Bilt, The Netherlands

[5]Aerospace Information Research Institute, Chinese Academy of Sciences (AirCAS), Beijing, 100101, China

[6]University of Mining and Technology (CUMT), School of Environment Science and Spatial Informatics, Xuzhou, 221116, China

[7]Hangzhou Meteorological Bureau, Hangzhou 310051, China

*Correspondence to:* Bin Zhu (binzhu@nuist.edu.cn) and Bu Yu (fengying5457@163.com)

**Abstract.**

Urban heat island (UHI) and urban air pollution are two major environmental problems faced by many metropolises. The UHI affects air pollution by changing the local circulation and the chemical reaction environment, e.g., air temperature and relative humidity. In this study, the WRF-CMAQ model was used to investigate the impact of an UHI on the vertical distribution of aerosol particles, especially secondary inorganic aerosol (SIA), taking the strong UHI in Hangzhou, China, as an example. The results show that during the day with the strongest UHI case of the 8-day simulation, the UHI effect resulted in a decrease of the $PM_{2.5}$ concentrations in the boundary layer (BL) by about 33%, accompanied by an increase in the lower free troposphere (LFT) by about 19%. This is mostly attributed to the UHI circulation (UHIC) effect, which accounted for 91% of the UHI-induced variations in $PM_{2.5}$, rather than the UHI temperature or humidity effects, which contributed only 5% and 4%, respectively. The UHIC effect plays a dominant role, ranging from 72% to 93%, in UHI-induced $PM_{2.5}$ variation in all 8 UHI cases. The UHIC not only directly transports aerosol particles from ground level to the LFT, but also redistributes aerosol precursors. During the strongest UHI case, about 80% of the UHIC-induced increase of the aerosol particles in the LFT is due to direct transport of aerosol particles, whereas the other 20% is due to secondary aerosol formation resulting from the transport of aerosol precursor gases. Of this 20%, 91% is contributed by secondary inorganic aerosol (SIA), especially ammonium nitrate aerosol formed from ammonia and nitric acid. In the atmosphere, ammonium nitrate is in equilibrium with ammonia and nitric acid and the equilibrium depends on the ambient temperature. In the lower urban BL, the temperature is higher than in the LFT and the ammonium nitrate equilibrium in the lower BL is more toward the gas phase than in the LFT; when these gases are transported by the UHIC into the colder LFT, the equilibrium shifts to the aerosol phase. Hence, the UHIC changes the vertical distribution of SIA, which may have potential implications on the radiation budget, cloud formation, and precipitation in the urban and surrounding areas.

## 1. Introduction

With the fast economic development, China has experienced rapid industrialization and urbanization in the past decades. The urbanization process has replaced natural land surface with artificial constructions. The radiative, aerodynamic, thermal and moisture properties of these urban constructions are radically different from the natural landscape. Lower albedo, larger energy storage, higher Bowen ratio, and the release of anthropogenic heat in urban areas, lead to the urban heat island (UHI) phenomenon (Rotach et al., 2005; Oke et al., 2017). Meanwhile, the increase in fossil fuel consumption and the rapid growth of the population have led to a sharp increase in anthropogenic emissions in urban areas, resulting in serious urban air pollution problems such as haze and photochemical smog (Zhao et al., 2017; Wang et al., 2019; Gao et al., 2017). Crutzen (2004) pointed out that the UHI and air pollution are far from being independent phenomena but co-existing.

Urbanization affects local meteorological conditions, which in turn influences the distribution and formation of air pollutants. UHI leads to strong thermal convection and diffusion, which promotes the vertical transport of heat, water vapor, and atmospheric pollutants from ground level to the upper atmosphere, and subsequently affects cloud formation, precipitation, and chemical reactions (Kang et al., 2014; Li et al., 2019; Zhu et al., 2015; Zhong et al., 2015). The vertical transport of water vapor caused by urban thermal convection is the source of water for cloud formation and precipitation (Baik et al., 2006), resulting in the increase of precipitation in the city and downwind (Changnon, 1979; Wan et al., 2013; Xie et al., 2016). Variations in cloud cover and precipitation will inevitably impact the liquid phase chemistry of aerosol and the wet removal of air pollutants.

The UHI effect also influences chemical processes in the urban atmosphere. The increase in air temperature accelerates certain chemical cycles in the atmosphere, most of which lead to enhanced ground-level ozone production (Narumi et al., 2009; Zhan and Xie, 2022). The UHI circulation (UHIC) can also affect chemical reactions by changing the distribution of air pollutants and their precursors. Zhu et al. (2015) indicated that the UHIC transports $NO_x$ and VOCs, the precursors of $O_3$, from ground level to higher altitudes, leading to a significant increase in the photochemical production of $O_3$ there. The impact of the UHI effect on $PM_{2.5}$ is very complicated. Liu et al. (2016) suggested that the UHIC and high surface roughness in urban areas reduce the horizontal wind speed at ground level, and hence suppress the outflow of air pollutants, resulting in the increase of $PM_{2.5}$. In contrast, other studies indicated that UHI reduces ground level aerosol concentrations by enhancing turbulent mixing in a deeper urban boundary layer (BL) (Zhu et al., 2017; Li et al., 2018; Liao et al., 2015). Several studies have investigated the impact of UHI on aerosol, but few of them focused on how UHI affects the secondary formation of inorganic aerosols, such as sulfate, nitrate, and ammonium. These scattering aerosols can attenuate incident solar radiation, leading to an overall cooling of the surface, inhibiting the development of the PBL (Li et al., 2017; Ma et al., 2020). In addition, sulfate and nitrate aerosol particles are very hygroscopic and thus act as cloud condensation nuclei in a high relative humidity environment (Tao et al., 2012). Therefore, the vertical distribution and secondary formation of inorganic aerosol in the urban environment plays an important role in the local radiation budget, cloud formation and precipitation.

In this study, a strong UHI case that occurred in Hangzhou on 18 September, 2017, was chosen to investigate the impact of the UHI effect on the distribution and formation of $PM_{2.5}$ and its inorganic components. We conducted four simulations using the regional meteorological model

WRF coupled with the chemical transport model CMAQ to simulate the UHI effect and quantify the impact of UHI-induced variations in circulation, temperature, and humidity on secondary inorganic aerosol (SIA). This paper is organized as follows. Section 2 describes the model configuration, experiment design, and verification. The UHI effect and its impact on $PM_{2.5}$ and its inorganic components are presented in section 3 and the conclusions in section 4.

## 2. Model Description and Verification

### 2.1 Model Description and Configuration

The WRF model version 3.9.1 coupled with a multilayer building energy parameterization (BEP) urban canopy model was employed to simulate the UHI effect and provide the meteorological fields for the chemical transport model. The model was set up with three nested domains (Fig. 1a) with horizontal grid numbers (grid spacing) of 300×300 (9 km), 240×240 (3 km), and 140×140 (1 km). The innermost domain was centered on Hangzhou city (Fig. 1b), the Yangtze River Delta's second largest megacity, located by Hangzhou Bay and with a population of over 12 million. The model was divided vertically into 37 sigma levels from the surface to 50 hPa, the lowest 20 of which are below 2 km to better resolve the processes within the BL and the 5 lowest layers, each with thickness of about 10 m, are within the urban canopy. The parameterization used in the WRF simulation is described in Table S1. A 10-day simulation (from 00:00 UTC on 10 September to 00:00 UTC on 20 September 2017) was conducted with the initial conditions (ICONs) and the outermost boundary conditions (BCONs) from the National Center for Environmental Prediction's $1^o$ grid spacing operational Global Forecast System final analyses. The fine resolution (30 s) Moderate Resolution Imaging Spectroradiometer (MODIS) 20 category land use data were used to represent the urban land type in the innermost domain. Two experiments were designed to investigate the UHI effect of Hangzhou city. The first, known as CTL simulation, used MODIS land use data as the default surface (with Hangzhou city) description for the control experiment; the second, NUB simulation, was a sensitivity experiment, in which the urban land use of Hangzhou city was replaced by cropland. By comparison of the result from the two experiments, the UHI effects of Hangzhou city can be identified. The 2-m air temperature difference between the two experiments (CTL−NUB) reflects the UHI intensity (UHII); the humidity difference indicates the intensity of urban dry island; and the difference in wind fields reveals the UHIC.

The CMAQ model version 5.0.2 was applied to simulate concentrations of gaseous and particulate air pollutants with the same grid settings as the WRF model. The output of the two WRF experiments (CTL and NUB) provided the meteorological inputs for the CMAQ simulations. The CMAQ simulations cover the same time period as the WRF simulations with the first 2 days used as spin-up time. The ICONs and BCONs for the outermost domain of the CMAQ simulations were derived from the Model for Ozone and Related Chemical Tracers version 4 (MOZART-4) (Emmons et al., 2010). The BCONs for the CMAQ nested domains were extracted from the immediate concentration files of their parent domains. The monthly Multi-resolution Emission Inventory for China (MEIC, http://www.meicmodel.org/, last accessed: 20 July 2021) with the reference year of 2017 was used for anthropogenic emissions except for Hangzhou. The anthropogenic emissions in Hangzhou were derived from the Hangzhou Municipal Ecology and Environment Bureau's anthropogenic emission survey in 2016 with a horizontal grid space of 1 km × 1 km. Biogenic emissions were generated from the Model for Emissions of Gases and

Aerosols from Nature (MEGAN) version 2.1 (Guenther et al., 2012). Sea-salt and dust emissions were calculated online. The CB05 and AERO6 mechanisms were chosen for gas-phase chemistry and aerosols, respectively.

The process analysis technique (Gipson, 1999), which can determine the contributions of the physical and chemical processes to atmospheric species, was implemented in the CMAQ
simulations. The processes discussed in this study include horizontal advection (HADV), vertical advection (ZADV), vertical diffusion (VDIF), and aerosol (AERO) processes.

It should be noted that all simulation experiments use the same emission inventory. Therefore, the impact of the UHI on aerosol properties can be derived from the difference between the CTL and NUB CMAQ simulations. In order to identify the impact of the UHI-induced changes in
temperature, humidity, and circulation (the UHIC is caused by the increased temperature in the urban area, but temperature and temperature-induced circulation affect air pollutants in different ways) on the aerosol properties, two additional sensitivity experiments were designed. One is the TMP simulation, in which the air temperature used in the aerosol module where the SIA formation was calculated, was derived from the NUB simulation, other parameters were kept the same as in
the CTL simulation. The other one is the T&H simulation, in which the air temperature and specific humidity were derived from the NUB simulation, while other parameters were kept the same as in the CTL simulation. The difference between CTL and TMP simulations indicates the UHI temperature effect on aerosol. By comparing the CTL and T&H simulations, the overall effect from temperature and humidity can be derived. Hence the humidity and circulation effects
can be separated. Details of the experimental designs are provided in table 1.

**Table 1.** Configuration of CMAQ experiments

| Experiment | Configuration | Effect |
|---|---|---|
| CTL | CTL meteorology from WRF. | |
| NUB | NUB meteorology from WRF. | CTL−NUB: Overall UHI effect on aerosol. |
| TMP | CTL meteorology but using NUB temperature in aerosol module. | CTL−TMP: UHI temperature effect on aerosol. |
| T&H | CTL meteorology but using NUB temperature and specific humidity in aerosol module. | CTL−T&H: UHI temperature effect + UHI humidity effect on aerosol. (CTL−T&H) −(CTL−TMP) = TMP−T&H: UHI humidity effect on aerosol. (CTL−NUB)−(CTL−T&H) = T&H−NUB: UHIC effect on aerosol. |

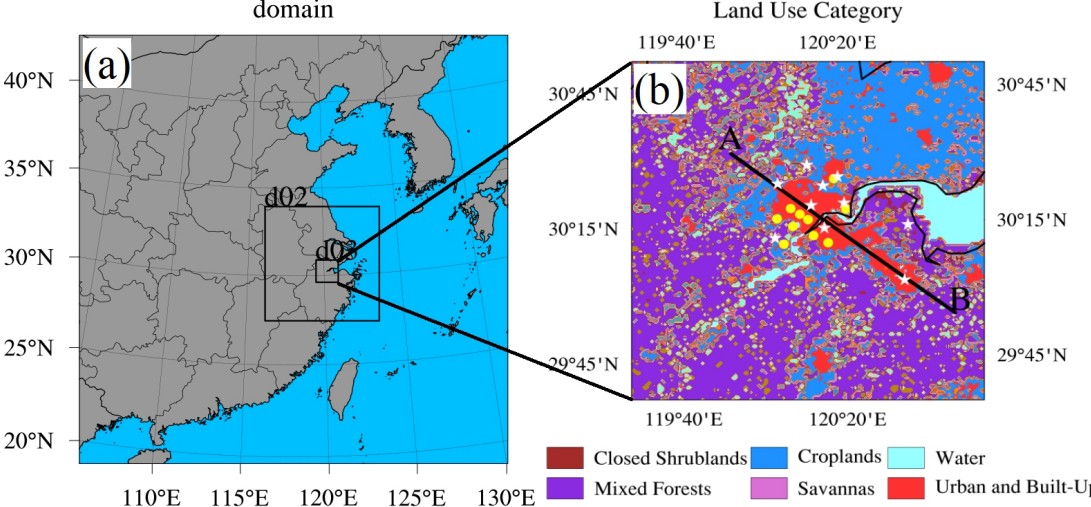

**Figure 1.** The three nested modeling domains (a) and land use categories over the innermost domain (b). The horizontal spacing for domain d01, d02, and d03 are 9 km, 3 km, and 1 km, respectively. The line AB in (b) denotes the location of the vertical cross section shown in Fig. 3 and Fig. 4. The yellow filled circles represent locations of the 10 $PM_{2.5}$ monitoring sites and the white stars are the locations of the 10 meteorological observation sites.

**2.2 Model Evaluation**

The model performance for the meteorological parameters (air temperature, relative humidity, and wind speed) and $PM_{2.5}$ concentrations in Hangzhou was extensively evaluated on an hourly scale, using surface measurements at 10 meteorological observation sites (white stars in Fig. 1b) and 10 $PM_{2.5}$ monitoring sites (yellow filled circles in Fig. 1b) released by the Hangzhou Meteorological Bureau (HMB) and the China National Environmental Monitoring Center (CNEMC), respectively. Statistical metrics including the correlation coefficient, normalized mean bias (NMB), normalized mean error (NME), index of agreement (IOA), mean fractional bias (MFB), and mean fractional error (MFE) were calculated to compare the model simulations with observations. The NMB, NME, IOA, MFB, and MFE were calculated as follows:

$$NMB = \frac{\sum_{i=1}^{N}(M_i - O_i)}{\sum_{i=1}^{N} O_i} \times 100\% \qquad (1)$$

$$NME = \frac{\sum_{i=1}^{N}|M_i - O_i|}{\sum_{i=1}^{N} O_i} \times 100\% \qquad (2)$$

$$IOA = 1 - \frac{\sum_{i=1}^{N}(M_i - O_i)^2}{\sum_{i=1}^{N}\left(|M_i - \overline{M}| + |O_i - \overline{O}|\right)^2} \qquad (3)$$

$$MFB = \frac{2}{N}\sum_{i=1}^{N}\left(\frac{M_i - O_i}{M_i + O_i}\right) \times 100\% \qquad (4)$$

$$MFE = \frac{2}{N}\sum_{i=1}^{N}\left(\frac{|M_i - O_i|}{M_i + O_i}\right) \times 100\% \qquad (5)$$

where $M_i$ are the hourly model simulations, $O_i$ are the hourly observations, and N denotes the number of data pairs.

The time series of hourly ground-level air temperature, relative humidity, wind speed, and $PM_{2.5}$ concentrations from both the CTL simulation and observations at the 10 HMB sites and the

10 CNEMC sites are compared in Fig. 2. The CTL simulation captures the diurnal variations and magnitude of meteorological parameters and $PM_{2.5}$ well at most of the sites. The correlation coefficient, NMB, and NME between observations and CTL simulations at the 10 meteorological observation sites and the 10 $PM_{2.5}$ monitoring sites are shown in Fig. 3. The correlation coefficients for air temperature and relative humidity are around 0.9 at all 10 sites, and for wind speed they are between 0.6 and 0.8 (Fig. 3a). The NMBs for air temperature and relative humidity at all sites are within ±10% and for wind speed they are within ±20%. The NMEs for air temperature, relative humidity, and wind speed at all 10 sites are <10%, <20%, and <50%, respectively (Fig. 3a). The correlation coefficients for $PM_{2.5}$ are >0.6 except for one site (Fig. 3b). The NMBs for $PM_{2.5}$ at all sites are within ±30%, and all NMEs are <60% (Fig. 3b). The IOA, MFB, and MFE for both meteorological parameters (Fig. S1a) and $PM_{2.5}$ (Fig. S1b) at each site show a pattern similar to that for the correlation coefficients, NMB, and NME, respectively. The average correlation coefficient, NMB, NME, IOA, MFB and MFE for $PM_{2.5}$ at the 10 sites are 0.68, 0.5%, 44.7%, 0.78, −12% and 45.4%, respectively. In general, the WRF-CMAQ model performed well in the simulation of both meteorological parameters and $PM_{2.5}$ concentrations.

To evaluate the impact of urban land surface on ground level $PM_{2.5}$ simulation, the NUB simulations were also compared with the observations. The results from these comparisons show that if the Hangzhou urban surface is replaced by cropland, the ground level $PM_{2.5}$ concentration would be larger by approximately 60% (NMB for the NUB experiment is 60%) and simulation error would increase by approximately 30% (NME for the NUB experiment is 74%). These results suggest that the urban land surface and UHI effect exert an important impact on urban air quality.

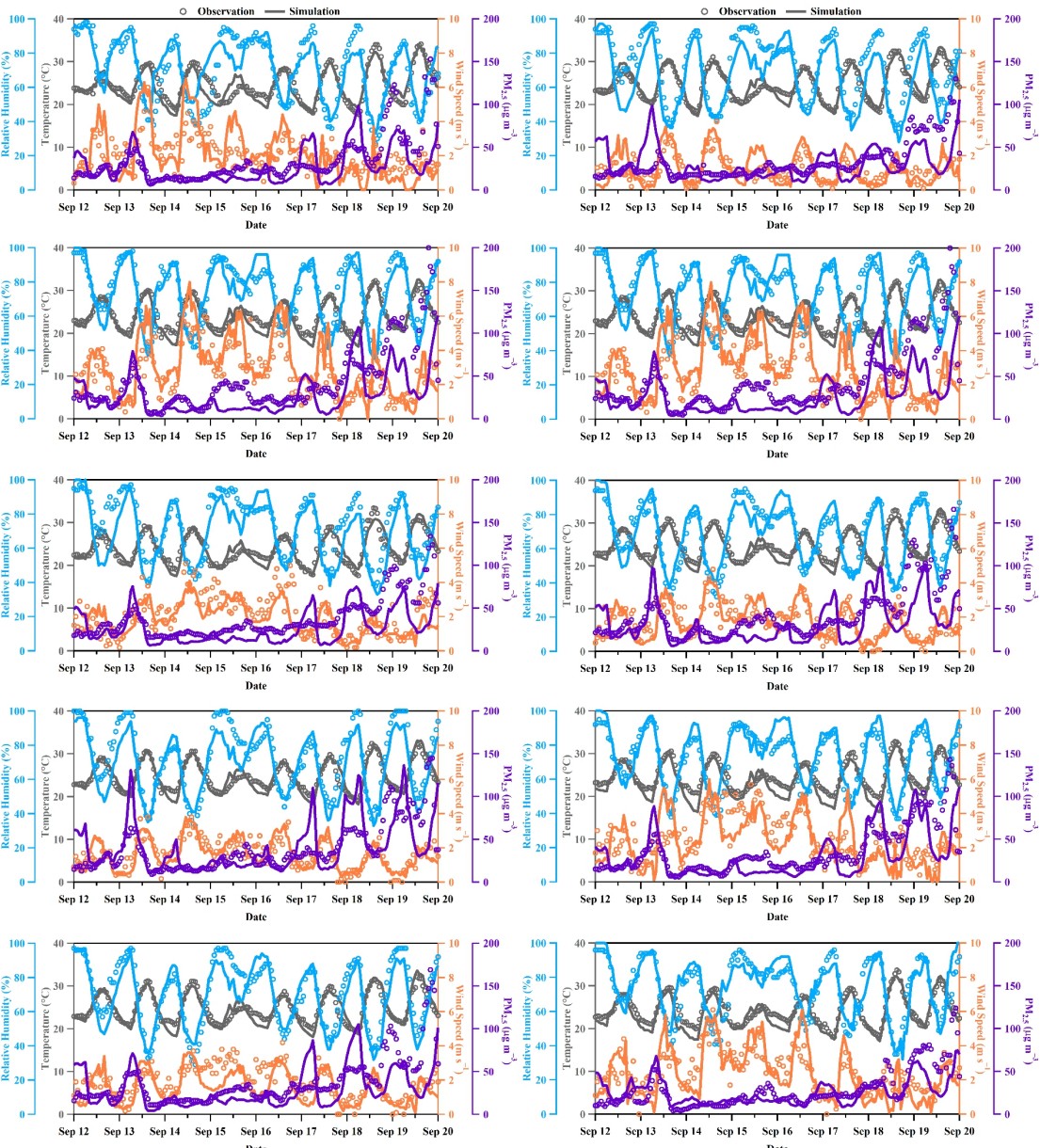

**Figure 2.** Time series of simulated and observed air temperature, relative humidity, wind speed, and PM2.5 concentration from 00:00 LT September 12 to 00:00 LT September 20, 2017 at 10 meteorological and 10 PM2.5 monitoring sites.

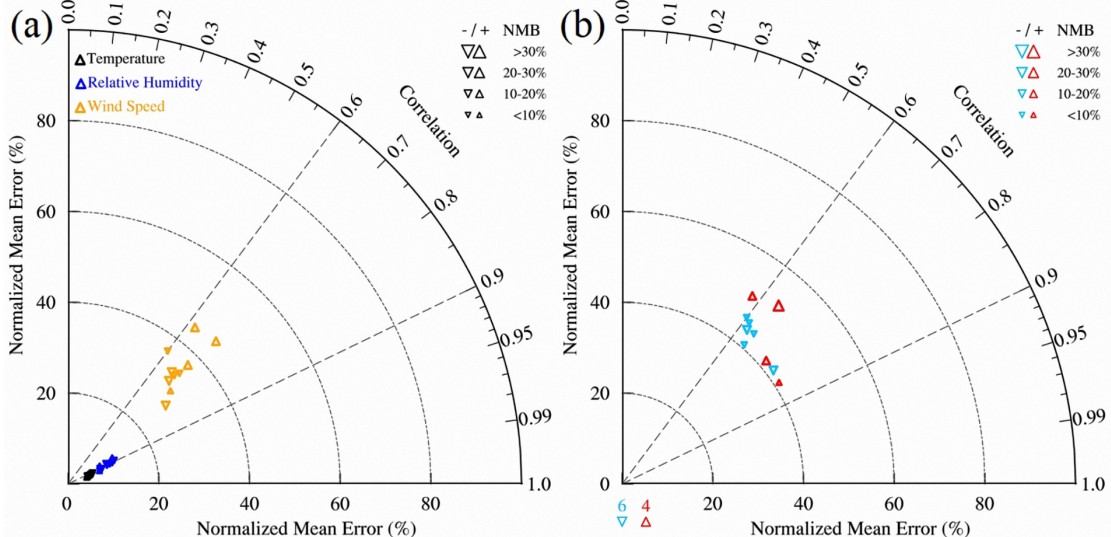

**Figure 3.** Taylor diagram of correlation coefficient, normalized mean bias (NMB) and normalized mean error (NME) between simulated (CTL experiment) and observed (a) air temperature, relative humidity, and wind speed at 10 sites in Hangzhou (white stars in Fig. 1b); and (b) $PM_{2.5}$ concentrations at 10 sites in Hangzhou (yellow filled circles in Fig. 1b).

## 3.  Results and Discussion

The UHI affects the concentrations of atmospheric pollutants and their spatial and vertical distribution by changing local circulation and chemical reaction environment, e.g., air temperature and humidity, in the overlying atmospheric column. The impacts of these effects over Hangzhou and its surrounding area were investigated by comparison of the meteorological parameters simulated in the CTL experiment with those from the NUB experiment.

### 3.1  Urban Heat Island Effect

The spatial-averaged (urban region of Hangzhou) UHII in each afternoon (12:00–17:00 LT) ranged from 1.1 °C to 1.9 °C during the 8-day (exclude 2 spin-up days) experiment. A strong UHI case is characterized by the low wind speed and high temperature. Low wind speeds are conductive to the coexistence of UHI with urban air pollution (Ulpiani, 2021). Observations show that $PM_{2.5}$ started to accumulate from 17 to 19 September due to decreasing wind speeds (Fig. 2). During the 8-day experiment, the strongest UHI case occurred on 18 September, which coincided with high temperatures and low wind speeds (Fig. 2). This case will be discussed in detail below.

The time-averaged (from 12:00 to 17:00 LT, on 18 September 2017) vertical distribution of air temperature, relative humidity, and wind vectors in the CTL experiment along line AB (see Fig. 1b) over Hangzhou show an obvious UHI effect, which is characterized by the higher air temperature, lower relative humidity, and stronger upward movement over the urban area than over the surrounding rural areas (Fig. S2). The differences in the air temperature, specific humidity and wind vectors between the CTL and NUB experiments along line AB show that the warming, drying, and circulation effects of the UHI reach an altitude of ~2.0 km (Fig. 4). Kang et al. (2014) suggested that the maximum surface UHII occurs in the evening, while the strongest UHIC appears in the afternoon. The UHIC reaches 2.0 km at 15:00–16:00 with a vertical speed of ~0.2 m s$^{-1}$, as shown by the diurnal variation of the UHII and UHIC profiles over Hangzhou (Fig. S3). Although the near-surface UHII is at its highest, the UHIC is insignificant in the evening.

The UHIC plays a prominent role in the horizontal and vertical distributions of air pollutants in urban atmosphere (Zhu et al., 2015). The changes in the air temperature and specific humidity may

potentially affect the secondary formation of aerosols. Meanwhile, the average BLH over Hangzhou in the CTL simulation was approximately 1.0 km, 0.3 km higher than that in NUB simulation. The deeper BL is the result of the stronger turbulence which is due to the warmer urban surface. As a result, the air pollutants are mixed over a deeper BL. The UHIC clearly shows the convergence in the lower BL and the strong upward transport except for a small area near the

urban center, where the Qiantang River is located. The vertical branches of UHIC penetrate the top of the urban BL (~1.0 km) to reach the lower free troposphere (LFT) (~2.0 km) (Fig. 4), which may have significant effects on the vertical transport of aerosol and its precursors. Similar UHI effects are also observed in other 7 cases (Fig. S4). The maximum height that the UHIC can reach varies from case to case, and is strongly dependent on the UHII and wind speed. In the 8-day

experiments, the difference in boundary-layer averaged vertical wind speed between the CTL and NUB experiments, which can be used to determine the intensity of UHIC, shows that when the UHII is strong and the BL wind speed is low, strong UHIC occurs, whereas when the UHII is weak and the BL wind speed is high, weak UHIC occurs (Table S2).

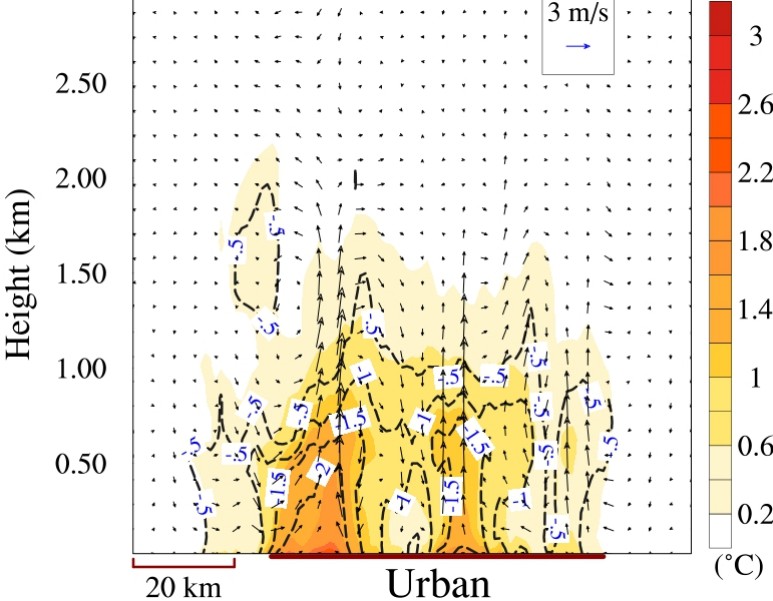

**Figure 4.** Vertical cross section of the difference in averaged (from 12:00 to 17:00 LT, on 18 September 2017) temperature (°C; filled colors), specific humidity (g kg$^{-1}$; dashed contour lines), and in-plane wind vectors (m s$^{-1}$; arrows) between the CTL and NUB experiments (CTL−NUB). Note that the vertical wind speed vectors are expanded by 10 times.

### 3.2 Impact of the UHI on PM$_{2.5}$

In the CTL experiment, the vertical distribution of PM$_{2.5}$ along the line AB shows higher PM$_{2.5}$ concentrations and a much deeper aerosol layer over the urban area than over the surrounding rural area (Fig. S5). To evaluate the UHI effect on the PM$_{2.5}$ vertical distribution, the vertical cross section of differences of the averaged (from 12:00 to 17:00 LT on 18 September 2017) PM$_{2.5}$ and in-plane wind vectors between the CTL and NUB experiments along the line AB

are shown in Fig. 5. The PM$_{2.5}$ differences show that PM$_{2.5}$ concentrations in the BL decreased, on average with 8.5 μg m$^{-3}$ (33%) and increased in the LFT, on average with 3.1 μg m$^{-3}$ (19%). The

other 7 cases show a similar effect of UHI on PM$_{2.5}$ (Fig. S6). However, the impact of UHI on PM$_{2.5}$ is negligible on 14 and 15 September, because on these two days the largest boundary-layer wind speed and the weakest UHI effect occurred (Table S2). Similar UHI effects were also obtained by WRF-Chem model simulations in the Yangtze River Delta (Liao et al., 2015), south China (Zhu et al., 2017) and California (Li et al., 2019).

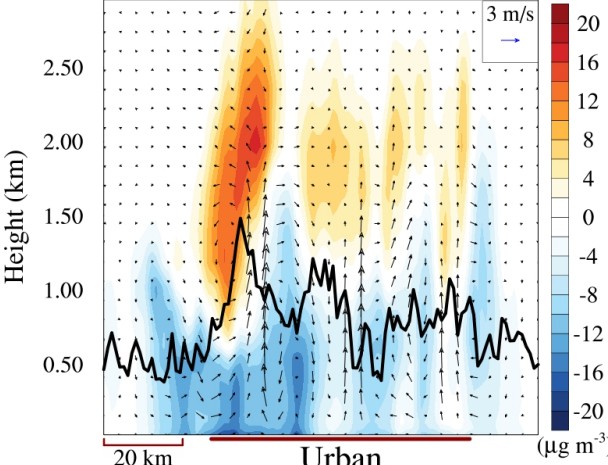

**Figure 5.** Vertical cross section of the differences in averaged (from 12:00 to 17:00 LT, on 18 September 2017) PM$_{2.5}$ concentration (μg m$^{-3}$; filled colors) and in-plane wind vectors (m s$^{-1}$; arrows) between the CTL and NUB experiments. Note that the vertical wind speed is expanded by a factor 10. The black line denotes the BL height.

In addition to influencing the distributions of the concentrations of aerosol particles and precursor gases by changing the local circulation, the UHI also affects the secondary formation of aerosol particles by changing the air temperature and the humidity. By comparing the four experiments (Table 1), the contributions of the changes in temperature, humidity, and circulation to the vertical distribution of the PM$_{2.5}$ concentrations over Hangzhou can be identified. Figure 6 shows the vertical profiles of the PM$_{2.5}$ concentrations from the CTL, NUB and T&H simulations and the effects of several processes on them. The difference between the PM$_{2.5}$ concentration profiles from the CTL and T&H simulations is very small as compared to the difference between the CTL and NUB simulations (left panel in Fig. 6. The PM$_{2.5}$ profile from the TMP simulation is not shown in the picture, because it is very close to the profiles from the CTL and T&H simulations.). The simulations show that the effects of air temperature and humidity on the PM$_{2.5}$ concentration account for only 9.7% of the total UHI effect, of which the contributions of temperature and humidity are 5% and 4%, respectively, while the contribution from the circulation effect accounts for about 91%.

To further investigate how the UHI affects the vertical distribution of PM$_{2.5}$ by changing physical and chemical processes, a process analysis technique was performed (Fig. 6). The profiles of the area-averaged PM$_{2.5}$ contribution from the vertical advection process (ZADV) show that in the CTL experiment the negative contributions (−25 μg m$^{-3}$ on average) in the middle and lower part of the BL are higher than in the NUB experiment (−5 μg m$^{-3}$ on average) whereas in the LFT the CTL contributions are positive (14 μg m$^{-3}$ on average) and stronger than in the NUB experiment (2 μg m$^{-3}$ on average). This suggests that the UHICs induce the transport of aerosol from the lower part of the BL to the upper part and into the free troposphere. The difference in horizontal advection process (HADV) between the CTL and NUB experiments indicates that the

295 UHIC results in stronger convergence (21 μg m$^{-3}$) of aerosol in the BL and stronger divergence (−15 μg m$^{-3}$) in the LFT.

The vertical diffusion process (VDIF) results in uniform mixing of PM$_{2.5}$ in the BL. In the afternoon of 18 September 2017, the mean BL height over Hangzhou was about 1 km. Stronger turbulent mixing in the CTL experiment due to higher temperature and urban surface roughness

resulted in a higher urban BL height and a smaller PM$_{2.5}$ vertical gradient than that in NUB experiment. Since the emissions of PM$_{2.5}$ are near the surface, there is a strong negative contribution from vertical diffusion near ground level (Fig. 6). The positive contributions in the middle and lower BL (<0.8 km) indicate that vertical diffusion transports aerosol from ground level to the upper atmosphere by turbulent mixing. Weak negative PM$_{2.5}$ contributions (~−1 μg

m$^{-3}$) from vertical diffusion, especially in CTL experiment, appears in upper BL and LFT.

The contributions of aerosol process (AERO) to PM$_{2.5}$ concentrations are negative (−3 μg m$^{-3}$ on average) in the lower BL (<0.4 km), but positive (4 μg m$^{-3}$ on average) in the upper air (0.4~2.0 km) (Fig. 6), which suggests that the aerosols are dissociated in the lower BL while they formed in the upper air. Under the influence of UHI, a much stronger secondary aerosol formation

occurs in the upper BL and the LFT. This is probably due to the transport of aerosol precursors from ground level to the upper air by the UHIC.

In the LFT, the UHIC contributed ~95% (the average contribution from ground level to LFT is ~90%) of the aerosol increase. Process analysis shows that vertical advection (ZADV) and aerosol (AERO) processes are responsible for this increase. The differences between the ZADV

and the AERO contributions from the T&H and NUB simulations show that direct vertical advection of aerosol contributes 80% of the UHIC-induced aerosol increase in the LFT, secondary formation of aerosol from vertically transported aerosol precursors contributes the remaining 20% of the UHIC effect.

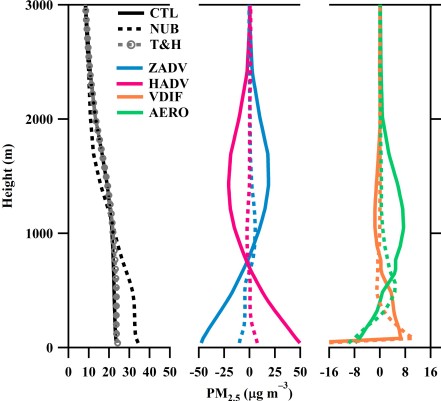

**Figure 6.** Vertical profiles of averaged (from 12:00 to 17:00 LT, on 18 September 2017) PM$_{2.5}$ concentrations from the CTL (solid line), NUB (dashed line) and T&H (dashed line and circle) experiments and contributions of vertical advection (ZADV), horizontal advection (HADV), vertical diffusion (VDIF), and aerosol (AERO) processes to PM$_{2.5}$ concentration from the CTL and NUB experiments.

**3.3 Impact of the UHI on Inorganic Aerosol**

To quantify the impact of the UHI on the dissociation and formation of SIA over the urban atmosphere, the process analysis technique was applied. The results in Fig.7 show that the UHI

affects the vertical distribution of sulfate mainly through circulation rather than secondary formation (Fig. 7a). The negative contributions of the aerosol process to PM$_{2.5}$ in the lower BL are mostly attributed to the dissociation of ammonium nitrate (Figs. 7b, 7c). Meanwhile, the positive contributions of the aerosol process in the upper air are mostly attributed to the secondary formation of ammonium nitrate aerosol. The formation and dissociation of ammonium nitrate aerosol involves an equilibrium reaction between the particle-phase *NH$_4$NO$_3$*, and gas-phase *HNO$_3$* and *NH$_3$*:

$$NH_4NO_3(s) \rightleftharpoons HNO_3(g) + NH_3(g). \quad (6)$$

Process analysis shows that the aerosol process provides positive contributions to ammonia and nitric acid in lower BL but negative contributions in the upper atmosphere (Fig. 8) - the opposite of the result from ammonium nitrate aerosol. This finding supports the conclusion that ammonium nitrate aerosols are formed in the upper atmosphere and dissociate in the lower BL.

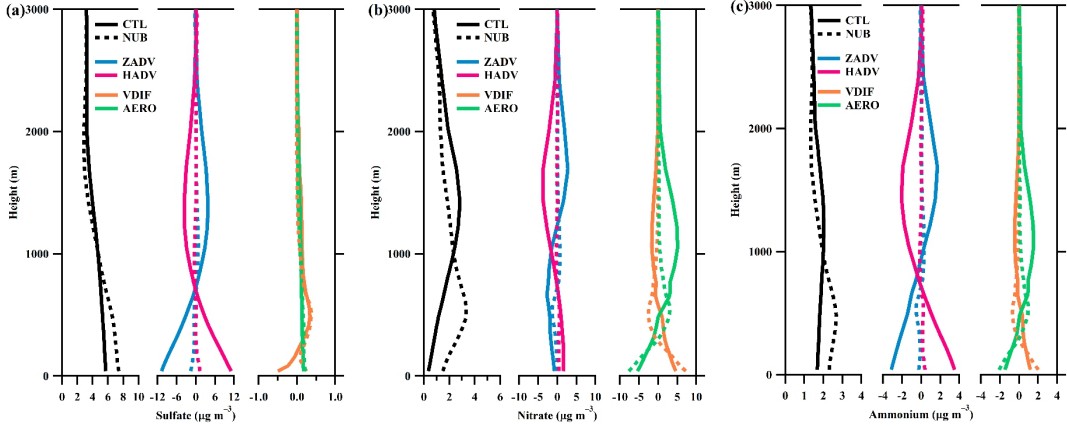

**Figure 7.** Averaged (from 12:00 to 17:00 LT, 18 September 2017) vertical profiles of sulfate (a) nitrate (b) and ammonium (c) aerosol concentrations and contributions of vertical advection (ZADV), horizontal advection (HADV), vertical diffusion (VDIF), and aerosol (AERO) processes to their mass concentrations in the CTL (solid line) and NUB (dashed line) experiments.

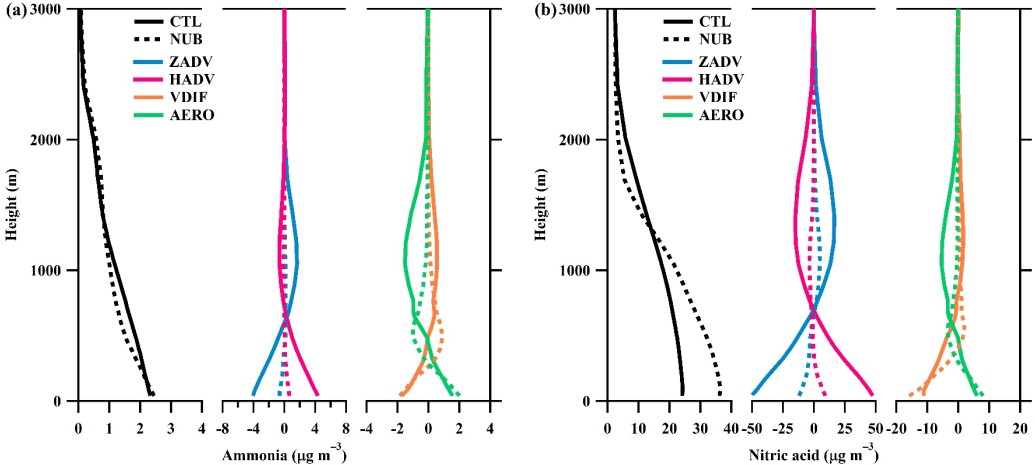

**Figure 8.** Averaged (from 12:00 to 17:00 LT, on 18 September 2017) vertical profiles of ammonia (a) and gas-phase nitric acid (b) concentrations and contributions of vertical advection (ZADV), horizontal advection (HADV), vertical diffusion (VDIF), and aerosol (AERO) processes to their mass concentrations in the CTL (solid line) and NUB (dashed line) experiments.

The dissociation of ammonium nitrate aerosol strongly depends on the air temperature, and

the dissociation constant in the model was calculated using equation 7 (Mozurkewich, 1993):

$$K = exp\left[118.87 - \frac{24084}{T} - 6.025ln(T)\right], \quad (7)$$

where $K$ (nanobar: $nb^2$) is the dissociation constant, $T$ (K) is atmospheric temperature. Air temperature over Hangzhou decreased from 30 °C at ground level to 20 °C at 1 km altitude, resulting in the decrease of the dissociation constant from 160 $nb^2$ near the surface to 10 $nb^2$ at 1 km. Therefore, in the warm lower urban BL, the ammonium nitrate equilibrium (eq. 6) shifts to dissociate into ammonia and nitric acid and at high elevation where the air is colder, the balance is on the side of ammonium nitrate aerosol. Consequently, the peak concentrations of the ammonium and nitrate aerosol profiles appear in the upper atmosphere rather than near ground level (Figs. 7b, 7c), which is consistent with flight observations in California (Neuman, 2003), Cabauw (Aan De Brugh et al., 2012), and model simulations in Milan (Curci et al., 2015). Through vertical diffusion, ammonium nitrate aerosol is transported from the upper air to the lower BL (Figs. 7b, 7c) where it dissociates into ammonia and nitric acid (Fig. 8). Meanwhile, ammonia and nitric acid are transported from ground level to the upper atmosphere by vertical advection and diffusion processes (Fig. 8) and then form ammonium nitrate aerosol.

For comparison with $NH_4NO_3$, the dissociation constant for the equilibrium relationship between particle-phase $(NH_4)_2SO_4$, and aqueous-phase $NH_4^+$ and $SO_4^{2-}$:

$(NH_4)_2SO_4(s) \rightleftharpoons 2NH_4^+(aq) + SO_4^{2-}(aq),$ (8)

was calculated using equation 9 (Kim et al., 1993):

$$K = K(T_0)\left[a\left(\frac{T_0}{T} - 1\right) + b\left(1 + ln\left(\frac{T_0}{T}\right) - \frac{T_0}{T}\right)\right], \quad (9)$$

where $T_0$ =298 K, $K(T_0)$ =1.817 $mol^3$ $kg^{-3}$, a=–2.65 and b=38.57. The dissociation constant of ammonium sulfate aerosol slowly decreased from 1.892 $mol^3$ $kg^{-3}$ near the surface to 1.736 $mol^3$ $kg^{-3}$ at 1 km. This is why there is less secondary formed sulfate in the LFT (Fig. 7a).

Comparison of the CTL and NUB experiments shows that the CTL experiment produces much more ammonium nitrate aerosol in the upper BL and LFT (Fig. 7b, 7c). This is due to the UHI effect, more ammonia and nitric acid gases and aerosol particles (provide reaction interface for ammonium nitrate) are transported from the surface to higher elevation (Fig. 8), and hence promote the secondary formation of ammonium nitrate aerosol in the cold upper atmosphere. The nitrate and ammonium aerosols formed in the LFT account for approximately 91% of the total secondary formed aerosol and 20% of the $PM_{2.5}$ at that altitude.

In general, the CTL experiment shows that the $PM_{2.5}$ concentration in the lower BL is reduced by the UHI effect, which can be attributed to the intensified upward transport of aerosol by strong vertical advection and turbulent mixing processes in the urban atmosphere. The larger $PM_{2.5}$ concentration in the LFT in the CTL simulation is ascribed to the inflow of aerosol by strengthened vertical advection and the secondary formation of inorganic aerosols. However, the impact of enhanced vertical advection and formation of SIA in the urban atmosphere are generally ignored in previous studies.

In the 8-day simulation period, the UHI effects decreased the $PM_{2.5}$ concentrations in the BL by 6% to 33%, and increased the $PM_{2.5}$ concentrations in the LFT by 1% to 19% (Table 2). The simulations show that the impact of the UHI on aerosol is highly dependent on the UHI effect's intensity and background wind speed. The impact of the UHI on $PM_{2.5}$ concentrations in both the BL and the LFT is larger on 18 and 19 September (Table 2), when the UHII and UHIC are

stronger and the BL wind speed is lower, than on other days (Table S2). On 14 September the BL wind speed was the highest and on 15 September the UHII was the weakest (Table S2), resulting in a substantially smaller UHI impact on the $PM_{2.5}$ concentrations on these days (Table 2). The UHIC effect consistently plays a dominant role (72% to 93%) in affecting the vertical distribution of $PM_{2.5}$ in all UHI cases (Table 2). During the 8-day experiment, the UHI effect increased the mean concentrations of $PM_{2.5}$, nitrate, ammonium, and sulfate in the LFT by 7%, 20%, 8%, and 3%, respectively (Table 2). This suggests that ammonium nitrate aerosols increased considerably in the LFT due to the UHI effects.

**Table 2.** Average $PM_{2.5}$ concentrations and the impact of the UHI effects on aerosol during the 8-day experiment, for each day between 12:00 and 17:00 LT.

| Date | PMBL[1] (µg m⁻³) | PMFT[2] (µg m⁻³) | C_PM_BL[3] (%) | C_PM_LFT[4] (%) | C_UHIC[5] (%) | C_UHIT[6] (%) | C_UHIH[7] (%) | C_N_LFT[8] (%) | C_A_LFT[9] (%) | C_S_LFT[10] (%) |
|---|---|---|---|---|---|---|---|---|---|---|
| Sep 12 | 20.1 | 14.5 | −7 | 4 | 78 | 14 | 8 | 11 | 4 | 0 |
| Sep 13 | 13.6 | 4.2 | −11 | 10 | 90 | 6 | 4 | 20 | 12 | 6 |
| Sep 14 | 9.6 | 5.6 | −6 | 1 | 93 | 5 | 2 | 4 | 1 | 0 |
| Sep 15 | 14.6 | 10.3 | −6 | 3 | 72 | 19 | 9 | 19 | 5 | −2 |
| Sep 16 | 12.7 | 6.8 | −9 | 5 | 89 | 7 | 4 | 15 | 9 | 3 |
| Sep 17 | 9.5 | 6.7 | −13 | 3 | 88 | 7 | 5 | 30 | 2 | 0 |
| Sep 18 | 26.0 | 16.1 | −33 | 19 | 91 | 5 | 4 | 33 | 18 | 11 |
| Sep 19 | 25.0 | 13.2 | −16 | 14 | 83 | 5 | 12 | 29 | 11 | 7 |

[1] Average $PM_{2.5}$ concentration in the urban BL in the CTL simulation. [2] Average $PM_{2.5}$ concentration in the LFT in the CTL simulation. [3] The proportion of the UHI-induced $PM_{2.5}$ variations in the urban BL. [4] The proportion of the UHI-induced $PM_{2.5}$ variations in the LFT. [5] Average contribution of the UHIC effect on the UHI-induced $PM_{2.5}$ variations. [6] Average contribution of the UHI temperature effect on the UHI-induced $PM_{2.5}$ variations. [7] Average contribution of the UHI humidity effect on the UHI-induced $PM_{2.5}$ variations. [8] The proportion of the UHI-induced nitrate aerosol variations in the LFT. [9] The proportion of the UHI-induced ammonium aerosol variations in the LFT. [10] The proportion of the UHI-induced sulfate variations in the LFT.

## 4. Conclusions

In this study, the impacts of the UHI on the transport, diffusion, formation, and distribution of $PM_{2.5}$, especially SIA, over Hangzhou city were investigated using different simulation experiments with the WRF-BEP-CMAQ model. By comparison of the results from the CTL and NUB experiments, the UHI effects were separated out. The result from the strongest UHI case (18 September 2017) during the 8-day simulation shows that in the afternoon, due to the UHI effect, the BLH increased by about 0.3 km, and the UHICs can penetrate the top of the BL to reach the LFT.

The UHI effect exerted profound impacts on the $PM_{2.5}$ concentrations over the urban area. In the absence of the UHI effect, the ground level $PM_{2.5}$ concentrations would be about 60% larger. In the strongest UHI case, the $PM_{2.5}$ concentration in the urban BL over Hangzhou decreased by ~33% due to the UHI effect, but in the LFT the $PM_{2.5}$ concentration increased by ~19%. This variation is mostly attributed to the effect of UHIC (accounting for 91%) rather than to the UHI temperature and humidity effects. The UHIC effect consistently plays a dominant role (72% to 93%) in affecting the vertical distribution of $PM_{2.5}$ in all UHI cases. Process analysis shows that UHIC strengthens vertical advection, resulting in the transport of a large amount of $PM_{2.5}$ and its precursors from ground level to the free troposphere. Elevated BL and strengthened turbulence caused by the UHI promotes vertical diffusion of $PM_{2.5}$, resulting in the decrease of $PM_{2.5}$ in the

430 lower BL.

Moreover, the UHI effect promoted the secondary formation of inorganic aerosols, especially nitrate and ammonium aerosols, in the upper BL and the LFT by transporting more precursors, e.g., ammonia and nitric acid, from ground level to the upper atmosphere. During the 8-day experiment, the UHI effect increased the mean concentrations of nitrate, ammonium, and sulfate in the LFT by
435 20%, 8%, and 3%, respectively. At higher altitudes, where the air temperature is substantially lower than at the surface, the dissociation constant for ammonium nitrate was reduced, nitrate was driven into the particle phase. In the strongest UHI case, the nitrate and ammonium aerosols formed in the upper BL and the LFT account for approximately 90% of the total secondary formed aerosol and 20% of $PM_{2.5}$ at that altitude. The UHI-induced SIA increase in the LFT may have
440 implications for the transmission of solar radiation, for cloud formation, and for precipitation in the urban and surrounding areas.

*Data availability*. The $PM_{2.5}$ observation data were obtained from a mirror of data from the CNEMC real-time publishing platform (https://quotsoft.net/air/). The MEIC anthropogenic emission data were acquired from http://meicmodel.org/. The hourly ground meteorological
parameters that used for model validation are available from https://zenodo.org/record/6939963#.YuPaGXZBzZt. Model outputs and 1 km × 1 km anthropogenic emission data in Hangzhou can be obtained from the authors upon request.

*Author contributions*. HK was responsible for paper writing, model simulation, and data analysis. BZ proposed the idea and designed the research. GL and RJA contributed to the paper revision and
450 language editing. BY designed the research and provided high resolution anthropogenic emission data. WL performed observation data collection.

*Competing interests*. The authors declare that they have no conflict of interest.

*Acknowledgements*. This work was supported by the National Natural Science Foundation of China (Grant No. 42021004 and No. 41605091). The study contributes to the ESA / MOST
455 cooperation project DRAGON5, Topic 3 Atmosphere, sub-topic 3.2 Air-Quality.

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
