# Peer review of "Impact of urban heat island on inorganic aerosol in the lower free troposphere: a case study in Hangzhou, China"

_Atmospheric Chemistry and Physics, 2022_

## Author Comment (AC1)

Abbreviations that will be used in response:
UHI: urban heat island
UHII: urban heat island intensity
UHIC: urban heat island circulation
BL: boundary layer
LFT: lower free troposphere

**Reviewer 1**

General comments

The article investigates the effect of the UHI circulation and the related temperature and humidity fields on the air pollution in the city of Hangzhou (China) using the modeling chain WRF-CMAQ. This topic is within the aim of ACP, however in my opinion the article does not present significant novel contributions. Concerning the general structure of the paper, the title is appropriate and coherent with the content. The overall presentation is well structured and clear, the introduction provides a comprehensive description of the scientific background of the work and the number and quality of the reference are appropriate. I think that authors should state more clearly the aim and the novelty of their work. Concerning the methodology, I think that a brief description of the study area is missing and could be appropriate. The numerical methods applied are well established, even if not exhaustively described (see "Specific comments"). Furthermore, in my opinion authors should: 1) delve into a more exhaustive analysis and discussion of the results, 2) extend the analysis of the results to a longer period (than a single day run), 3) consequently enhance the conclusions. For example, the role of the wind speed intensity is just mentioned only at the end of the Results section.

**Response:** Thank you for your comments.

The aim of this study is to find out how UHI effects affect the vertical distribution of aerosol, particularly inorganic aerosol.

Novelty: In this study, we separated and quantified the impacts of the UHI temperature, humidity, and circulation effects on $PM_{2.5}$, and we found that the UHIC plays a dominant role, ranging from 72% to 93%, in the decrease/increase of $PM_{2.5}$ in the BL/LFT in all 8 UHI cases. We also discovered that the UHIC transports more ammonia and nitrate acid from the warm BL to the cold LFT, promoting the secondary formation of ammonium nitrate in the LFT, accounting for more than 90% of the secondary formed aerosol at that altitude.

A brief description of the study area is added to line 96 in section 2.1.

In this revised version, a more comprehensive and exhaustive analysis and discussion of the results are presented. More details about the UHI effect and its impact on $PM_{2.5}$ in the other 7 cases are provided (Figs. S4, S6, Table S2, Table 2). The UHI effects were found to be widespread in all 8-day experiments, and the impact of UHI on $PM_{2.5}$ is strongly dependent on the intensity of UHI effects and the boundary-layer wind speed. Stronger UHII and UHIC, as well as lower BL wind speed, are favorable for the increase of aerosol in the LFT. The details of the relationship among UHII, UHIC, boundary-layer wind speed and their impact on aerosol in the BL and the LFT are listed in Table S2 and Table 2.

**Specific comments**

Abstract:

**C1.** Lines 25-26: "This is mostly attributed to the UHI circulation (~90%) rather than changes in the air temperature (~5%) and humidity (~4%)." In my opinion this sentence does not help the reader to understand the results in this context.

**Response:** Thank you for the comments. We changed this sentence to "This is mostly attributed to the UHI circulation (UHIC) effect, which accounted for 91% of the UHI-induced variations in $PM_{2.5}$, rather than the UHI temperature or humidity effects, which contributed only 5% and 4%, respectively.".

Methodology:

**C2.** Line 88: Why a dated version of the WRF model (i.e., 3.9.1) has been used? Can the authors motivate this choice?

**Response:** Thank you for your comments. We chose WRF version 3.9.1 because it has been extensively evaluated in our previous urban heat island simulation studies and it is easier to couple with CMAQ version 5.0.2 than the latest released WRF version. The simulated meteorological parameters show good agreement with surface observations (model evaluation for meteorological parameters are added in section 2), which indicates that the WRF 3.9.1 can be used to investigate the urban heat island effects in this study. In the future, we will try the newly updated and extensively evaluated versions of WRF and CMAQ.

**C3.** Line 94: I have some doubts about the vertical resolution. As is known, BEP needs a high resolution near the surface to be used effectively. The sentence "the lowest 20 of which are below 2 km to better resolve the processes within the BL" does not provide enough information on the resolution near the soil. Furthermore, in my opinion a more detailed description of the physics options in the modeling chain (WRF-CMAQ) could be provided to the reader, maybe in the supplement. For example, which PBL option (in WRF) has been applied?

**Response:** Thank you for your valuable suggestion. The resolution of the vertical levels in the urban canopy layer is typically ~10 m. In this study, the urban canopy layer (from the surface to ~50 m) is divided into 5 levels. The sentence "The model was divided vertically into 37 sigma levels, the lowest 20 of which are below 2 km to better resolve the processes within the BL." was changed to "The model was divided vertically into 37 sigma levels from the surface to 50 hPa, the lowest 20 of which are below 2 km to better resolve the processes within the BL and the 5 lowest layers, each with thickness of about 10 m, are within the urban canopy.".

The parameterization used in WRF-CMAQ simulation is described in Table S1.

Table S1. Parameterizations used in WRF-CMAQ simulations

| | WRF | | CMAQ |
|---|---|---|---|
| Microphysics | Thompson scheme | Gas chemistry scheme | CB05 |
| Short- and long-wave radiation | Rapid Radiative Transfer Model for General circulation models (RRTMG) scheme | Aerosol scheme | Aero6 |
| PBL scheme | Bougeault-Lacarrère PBL (BouLac) | | |
| Land surface model | Four-layer Noah Land Surface Model | | |

**C4.** Lines 94-95: "A 10-day simulation (from 95 00:00 UTC 10 September to 00:00 UTC 20

September 2017) was conducted". A focused investigation on the UHI circulation should be performed during a period not affected by other local phenomena, such as the sea-land breezes or a heat wave. Why this period has been simulated? Can the authors explain if the absence of other local circulations has been verified? And if so, how? Furthermore, Fig. 2 shows an increasing trend in the $PM_{2.5}$ concentration compared to the previous days starting precisely from 17-18 September. Can authors comment on this?

**Response:** Thank you for your comments.

We chose this period because a strong and typical UHI case, which is characterized by low wind speed and clear weather (Fig. 2), occurred on September 18. The UHI effect and its impact on $PM_{2.5}$ was assessed in all 8 cases, but only the strongest UHI case was discussed in detail in the paper.

Hangzhou city is located by the sea, it is affected by sea-land breezes. The intensity of the sea-land breeze, background meteorology, and other local circulations all affect the UHII, e.g., a strong sea-land breeze or background wind may result in weak UHII. As we listed in Table S2 in the supplementary file, due to changes in the background wind and local circulations, the UHI effect's intensity varied from case to case during the 8-day experiment. Strong boundary-layer wind speed resulted in a weak UHI effect (September 14 and 15), whereas low boundary-layer wind speed caused a strong UHI effect (September 18 and 19). Therefore, we chose the strongest UHI case (September 18), which was less affected by background weather and other local circulations, to discuss its impact on aerosol particles in detail. Furthermore, the difference in wind vectors between the CTL and NUB experiments can be used to rule out effects of background wind and other local circulations.

The increasing trend in $PM_{2.5}$ from 17-18 September is due to the lower wind speed compared to previous days, and hence the $PM_{2.5}$ started to accumulate. We added the sentence "Low wind speeds are conductive to the coexistence of UHI with urban air pollution (Ulpiani, 2021). Observations show that $PM_{2.5}$ started to accumulate from 17 to 19 September due to decreasing wind speeds (Fig. 2)." to line 213.

**C5.** Model evaluation: Can the authors explain why the evaluation is performed only in terms of $PM_{2.5}$? Temperature and humidity are key parameters in this analysis; therefore I would expect the validation of the model also in terms of these variables, together with the wind speed.

**Response:** Thank you for your suggestion. The model evaluation in terms of temperature, relative humidity, and wind speed has now been added in section 2.2. The time series of hourly ground-level air temperature, relative humidity, and wind speed from both the CTL simulation and observations at the 10 Hangzhou Meteorological Bureau sites are compared in Fig. 2. The correlation coefficient, NMB, and NME between observations and CTL simulations at the 10 meteorological observation sites are shown in Fig. 3a. The IOA, MFB, and MFE for the three meteorological parameters are shown in Fig. S1a). Details of the description of model evaluation are given in section 2.2. In general, the model performed well in the simulation of both meteorological parameters and $PM_{2.5}$ concentrations, as shown in the following Figures (added to the text).

[Figure]

**Figure 2.** Time series of simulated and observed air temperature, relative humidity, wind speed, and PM$_{2.5}$ concentration from 00:00 LT September 12 to 00:00 LT September 20, 2017 at 10 meteorological and 10 PM$_{2.5}$ monitoring sites.

[Figure]

**Figure 3.** Taylor diagram of correlation coefficient, normalized mean bias (NMB) and normalized mean error (NME) between simulated (CTL experiment) and observed (a) air temperature, relative humidity, and wind speed at 10 sites in Hangzhou (white stars in Fig. 1b); and (b) $PM_{2.5}$ concentrations at 10 sites in Hangzhou (yellow filled circles in Fig. 1b).

[Figure]

**Figure S1.** Taylor diagram of index of agreement, mean fraction bias (MFB) and mean fraction error (MFE) between simulated (CTL experiment) and observed (a) air temperature, relative humidity, and wind speed at 10 sites in Hangzhou (white stars in Fig. 1b); and (b) $PM_{2.5}$ concentrations at 10 sites in Hangzhou (yellow filled circles in Fig. 1b).

Results and discussion:

**C6.** Lines 184-185: "In the 10-day experiment, the strongest UHI effect occurred on September 18, 2017, which will be discussed in the following sections". In my opinion a single day run does not provide sufficient data for a high quality analysis. Also, how is UHI quantified? How was the UHI on September 18th found to be the strongest over the 10-day period? Please clarify these aspects.

**Response:** Thank you for your comments. In this study, the strongest UHI effect was discussed in detail to emphasize the impact of UHI effect on aerosol. However, that does not mean that the impact of UHI on aerosol only exists on that specific day. The results from the 8-day experiment

show that the UHI effect is evident in other UHI cases and it is highly dependent on the intensity of the UHI effect.

The UHII is quantified by the air temperature difference between the CTL (with Hangzhou urban land surface) and NUB (without Hangzhou urban land surface) experiments. The sentence "The 2-m air temperature difference between the two experiments (CTL−NUB) reflects the UHI intensity (UHII); the humidity difference indicates the intensity of urban dry island; and the difference of wind fields reveals the UHIC." is added to line 111 in section 2.1.

By comparing the temperature difference (CTL-NUB) in the 8-day simulation, we found that the UHII is strongest on 18 September. The text "The spatial-averaged (urban region of Hangzhou) UHII in each afternoon (12:00–17:00 LT) ranged from 1.1 °C to 1.9 °C during the 8-day (exclude 2 spin-up days) experiment. Strong UHI case is characterized by the low wind speed and high temperature. Low wind speeds are conductive to the coexistence of UHI with urban air pollution (Ulpiani, 2021). Observations show that $PM_{2.5}$ started to accumulate from 17 to 19 September due to decreasing wind speeds (Fig. 2). During the 8-day experiment, the strongest UHI case occurred on 18 September, which coincided with high temperatures and low wind speeds (Fig. 2). This case will be discussed in detail below." is added to line 211-217 in section 3.1.

**C7.** Lines 193-195: "Kang et al. (2014) suggested that the maximum surface UHI intensity occurs in the evening, while the strongest UHI circulation appears in the afternoon." I think that some comments on the temporal characterization of the UHI circulation in the present results might also be interesting.

**Response:** Thank you for your valuable suggestion. We plotted the diurnal variation of UHII and UHIC and added it to the supplementary file (Fig. S3). The following sentences have been added to line 226 in section 3.1: The UHIC reaches 2.0 km at 15:00–16:00 with a vertical speed of ~0.2 m s$^{-1}$, as shown by the diurnal variation of the UHII and UHIC profiles over Hangzhou (Fig. S3). Although the near-surface UHII is at its highest, the UHIC is insignificant in the evening.

[Figure]

**Figure S3.** Spatially averaged diurnal variation of the differences between the vertical profiles from the CTL and NUB experiments over Hangzhou on September 18, 2017, for temperature (filled colors), specific humidity (contour lines), and vertical speed (arrows; note that the vertical speed is expanded by 10 times).

**C8.** Lines 337-338: "In the 10-day simulation period, 7 UHI cases occurred in Hangzhou. The average (12:00−17:00 LT each day) UHI intensity of these cases varied from 1.4 °C to 1.9 °C.". I have some concerns about the position of this sentence at the end of the Results section. Furthermore, could the authors comment the occurrence of 7 UHI cases in the 10-day period considered? It could be interesting to clarify what happened in the remaining three days and how the UHI intensity was computed.

**Response:** Thank you for your comments. We made a mistake in the previous version, the UHI effects occur in all 8 simulation days (exclude 2 spin-up days) (Table S2). The following discussion of the UHII in the 8-day period was added to lines 211-214 and lines 238-244.

Lines 211-214: The spatial-averaged (urban region of Hangzhou) UHII in each afternoon (12:00–17:00 LT) ranged from 1.1 °C to 1.9 °C during the 8-day (exclude 2 spin-up days) experiment. A strong UHI case is characterized by the low wind speed and high temperature. Low wind speeds are conductive to the coexistence of UHI with urban air pollution (Ulpiani, 2021).

Lines 238-244: Similar UHI effects are also observed in other 7 cases (Fig. S4). The maximum height that the UHIC can reach varies from case to case, and is strongly dependent on the UHII and wind speed. In the 8-day experiments, the difference in vertical wind speed between the CTL and NUB experiments, which can be used to determine the intensity of UHIC, shows that when the UHII is strong and the BL wind speed is low, strong UHIC occurs, whereas when the UHII is weak and the BL wind speed is high, weak UHIC occurs (Table S2).

The UHII is quantified by the air temperature difference between the CTL (with Hangzhou urban land surface) and NUB (without Hangzhou urban land surface) experiments. The sentence "The 2-m air temperature difference between the two experiments (CTL–NUB) reflects the UHI intensity (UHII); the humidity difference indicates the intensity of urban dry island; and the difference in wind fields reveals the UHIC." is added to line 111 in section 2.1.

**C9.** Lines 338-339: "the UHI circulation, which is characterized by vertical wind speed, is strengthened by 0 cm s−1 to 10 cm s−1 in the BL and 0 cm s−1 to 5 cm s−1 in the LFT". This sentence is not clear. Please rephrase. Moreover, please clarify the type of speed values. Are they maximum or average values over the 10-day period? Or what time do they refer to? Can authors please provide more details on this?

**Response:** Thank you for your suggestion. This sentence has been deleted. Instead, we listed the average boundary-layer wind speed, average boundary-layer vertical wind speed difference between the CTL and NUB experiments, and average vertical wind speed difference between the CTL and NUB experiments in the LFT during the afternoon (12:00–17:00 LT each day) in Table S2 in the supplementary file. Discussion about the relationship between wind speed and the UHIC was added to lines 240-244.

Lines 240-244: In the 8-day experiments, the difference in boundary-layer averaged vertical wind speed between the CTL and NUB experiments, which can be used to determine the intensity of UHIC, shows that when the UHII is strong and the BL wind speed is low, strong UHIC occurs, whereas when the UHII is weak and the BL wind speed is high, the UHIC is weak (Table S2).

Table S2. Average UHI effects during the 8-day experiment, for each day between 12:00 and 17:00 LT.

| Date | UHII (°C) | WS_BL[1] (m s⁻¹) | W_BL_D[2] (cm s⁻¹) | W_FT_D[3] (cm s⁻¹) |
|---|---|---|---|---|
| **Sep 12** | 1.4 | 4.5 | 1.2 | 1.0 |
| **Sep 13** | 1.6 | 6.9 | 1.1 | 0.6 |
| **Sep 14** | 1.6 | 8.3 | −0.2 | 0.0 |
| **Sep 15** | 1.1 | 7.8 | 0.2 | −0.5 |
| **Sep 16** | 1.4 | 6.6 | 1.1 | 0.6 |
| **Sep 17** | 1.5 | 4.3 | 0.3 | 1.9 |
| **Sep 18** | 1.9 | 2.1 | 10.0 | 5.9 |
| **Sep 19** | 1.6 | 2.7 | 6.4 | 2.5 |

[1] Averaged horizontal wind speed in the urban BL of Hangzhou. [2] Difference of averaged boundary-layer vertical velocity between the CTL and NUB simulations. [3] Difference of averaged vertical velocity in the LFT between the CTL and NUB simulations.

**C10.** Lines 340-341: "These UHI effects decreased the $PM_{2.5}$ concentrations in the BL by 1% to 26%, and increased the $PM_{2.5}$ concentrations in the LFT by 5% to 21%." As for the point above, it might be useful to have more details on the listed $PM_{2.5}$ values.

**Response:** Thank you for your suggestion. Details on the average $PM_{2.5}$ concentrations in the BL and the LFT as well as the contributions of the UHI effect on $PM_{2.5}$ in both the BL and the LFT are listed in Table 2. And we changed this sentence to "In the 8-day simulation period, the UHI effects decreased the $PM_{2.5}$ concentrations in the BL by 6% to 33%, and increased the $PM_{2.5}$ concentrations in the LFT by 1% to 19% (Table 2).". In the previous version, the average $PM_{2.5}$ concentrations and those contributions were calculated based on the vertical cross-section (Fig. 5). Instead, in this revised version, we calculated the average $PM_{2.5}$ concentrations and those contributions for the entire Hangzhou urban area. As a result, the contributions in Table 2 show little difference from those in the previous version.

**Table 2.** Average $PM_{2.5}$ concentrations and the impact of the UHI effects on aerosol during the 8-day experiment, for each day between 12:00 and 17:00 LT.

| Date | PMBL[1] (μg m⁻³) | PMFT[2] (μg m⁻³) | C_PM_BL[3] (%) | C_PM_LFT[4] (%) | C_UHIC[5] (%) | C_UHIT[6] (%) | C_UHIH[7] (%) | C_N_LFT[8] (%) | C_A_LFT[9] (%) | C_S_LFT[10] (%) |
|---|---|---|---|---|---|---|---|---|---|---|
| Sep 12 | 20.1 | 14.5 | −7 | 4 | 78 | 14 | 8 | 11 | 4 | 0 |
| Sep 13 | 13.6 | 4.2 | −11 | 10 | 90 | 6 | 4 | 20 | 12 | 6 |
| Sep 14 | 9.6 | 5.6 | −6 | 1 | 93 | 5 | 2 | 4 | 1 | 0 |
| Sep 15 | 14.6 | 10.3 | −6 | 3 | 72 | 19 | 9 | 19 | 5 | -2 |
| Sep 16 | 12.7 | 6.8 | −9 | 5 | 89 | 7 | 4 | 15 | 9 | 3 |
| Sep 17 | 9.5 | 6.7 | −13 | 3 | 88 | 7 | 5 | 30 | 2 | 0 |
| Sep 18 | 26.0 | 16.1 | −33 | 19 | 91 | 5 | 4 | 33 | 18 | 11 |
| Sep 19 | 25.0 | 13.2 | −16 | 14 | 83 | 5 | 12 | 29 | 11 | 7 |

[1] Average $PM_{2.5}$ concentration in the urban BL in the CTL simulation. [2] Average $PM_{2.5}$ concentration in the LFT in the CTL simulation. [3] The proportion of the UHI-induced $PM_{2.5}$ variations in the urban BL. [4] The proportion of the UHI-induced $PM_{2.5}$ variations in the LFT. [5]

Average contribution of the UHIC effect on the UHI-induced $PM_{2.5}$ variations. [6] Average contribution of the UHI temperature effect on the UHI-induced $PM_{2.5}$ variations. [7] Average contribution of the UHI humidity effect on the UHI-induced $PM_{2.5}$ variations. [8] The proportion of the UHI-induced nitrate aerosol variations in the LFT. [9] The proportion of the UHI-induced ammonium aerosol variations in the LFT. [10] The proportion of the UHI-induced sulfate variations in the LFT.

**C11.** Lines 342-343: "The result of the simulations show that the impact of the UHI on aerosol is highly dependent on the intensity of the UHI effect." The resultS show… Furthermore, this sentence is a bit to general. Could authors rephrase it?

**Response:** Thank you for your suggestion. We changed this sentence to "The simulations show that the impact of the UHI on aerosol is strongly dependent on the UHI effect's intensity and background wind speed.", and we used the following two examples to illustrate how the UHI effect's intensity and background wind speed affect aerosol "The impact of the UHI on $PM_{2.5}$ concentrations in both the BL and the LFT is larger on 18 and 19 September (Table 2), when the UHII and UHIC are stronger and the BL wind speed is lower, than on other days (Table S2). On 14 September the BL wind speed was the highest and on 15 September the UHII was the weakest (Table S2), resulting in a substantially smaller UHI impact on the $PM_{2.5}$ concentrations on these days (Table 2).". These sentences were added to lines 381-387.

---

## Author Comment (AC2)

Abbreviations that will be used in response:

UHI: urban heat island

UHII: urban heat island intensity

UHIC: urban heat island circulation

BL: boundary layer

LFT: lower free troposphere

**Reviewer 2**

Aerosol pollution is of great concern in many megacities all over the world. It is meaningful to study how urbanization affects local meteorology and air pollution. By using WRF-CMAQ model, this manuscript assessed the effects of UHI on the distribution and formation of aerosols in urban atmosphere. The authors separated the impacts of UHI circulation, temperature and humidity, and quantified the contributions of them on inorganic aerosols in the lower free troposphere. Generally, this manuscript is well organized. The findings are very interesting, the discussion is of scientific meaning, and the results can improve the understanding of the formation of inorganic aerosols in urban areas. This manuscript could be considered for potential publication after the following minor revisions.

1) In 2.2, it is better to show the results of the comprehensive model validation for meteorological parameters, such as temperature, relative humidity and wind speed.

**Response:** Thank you for this suggestion. The model evaluation in terms of temperature, relative humidity, and wind speed has now been added in section 2.2. The time series of hourly ground-level air temperature, relative humidity, and wind speed from both the CTL simulation and observations at the 10 Hangzhou Meteorological Bureau sites are compared in Fig. 2. The correlation coefficient, NMB, and NME between observations and CTL simulations at the 10 meteorological observation sites are shown in Fig. 3a. The IOA, MFB, and MFE for the three meteorological parameters are shown in Fig. S1a). Details of the description of model evaluation are given in section 2.2. In general, the model performed well in the simulation of both meteorological parameters and $PM_{2.5}$ concentrations, as shown in the following Figures (added to the text).

[Figure]

**Figure 2.** Time series of simulated and observed air temperature, relative humidity, wind speed, and PM₂.₅ concentration from 00:00 LT September 12 to 00:00 LT September 20, 2017 at 10 meteorological and 10 PM₂.₅ monitoring sites.

[Figure]

**Figure 3.** Taylor diagram of correlation coefficient, normalized mean bias (NMB) and normalized mean error (NME) between simulated (CTL experiment) and observed (a) air temperature, relative humidity, and wind speed at 10 sites in Hangzhou (white stars in Fig. 1b); and (b) PM$_{2.5}$ concentrations at 10 sites in Hangzhou (yellow filled circles in Fig. 1b).

[Figure]

**Figure S1.** Taylor diagram of index of agreement, mean fraction bias (MFB) and mean fraction error (MFE) between simulated (CTL experiment) and observed (a) air temperature, relative humidity, and wind speed at 10 sites in Hangzhou (white stars in Fig. 1b); and (b) PM$_{2.5}$ concentrations at 10 sites in Hangzhou (yellow filled circles in Fig. 1b).

2) Line 145: "The NMB, NME, MFB, and MFE…" should be "The NMB, NME, IOA, MFB, and MFE…".

**Response:** Yes, you are right, thank you for catching this omission. We added the "IOA" here now.

3) In 3.2, the process analysis technique was used, but it was not introduced in methodology. It is better to briefly describe it in section 2.

**Response:** Thank you for your suggestion. We added the description of the process analysis technique to line 129 in section 2 as follows.

The process analysis technique (Gipson, 1999), which can determine the contributions of the physical and chemical processes to atmospheric species, was implemented in the CMAQ simulations. The processes discussed in this study include horizontal advection (HADV), vertical advection (ZADV), vertical diffusion (VDIF), and aerosol (AERO) processes.

4) Lines 337-343: The authors should demonstrate that the conclusions (about the impact of UHI on inorganic aerosol in the lower free troposphere) are universal rather than just in a specific case. Although other 7 UHI cases in the 10-day simulation period were mentioned in the last paragraph, the discussion was not clear. More details about the 7 UHI cases should be provided in section 3.

**Response:** Thank you for your valuable suggestion. Details about the impact of UHI on aerosol ant its inorganic compositions during the 8-day simulation period (exclude 2 spin-up days) are listed in Table S2. From Table S2, we can find that the impact of the UHI on inorganic aerosols, particularly on nitrate and ammonium aerosols, is evident in all 8 UHI cases. The following detailed discussion has been added to lines 380-392.

Lines 380-392: In the 8-day simulation period, the UHI effects decreased the $PM_{2.5}$ concentrations in the BL by 6% to 33%, and increased the $PM_{2.5}$ concentrations in the LFT by 1% to 19% (Table 2). The simulations show that the impact of the UHI on aerosol is highly dependent on the UHI effect's intensity and background wind speed. The impact of the UHI on $PM_{2.5}$ concentrations in both the BL and the LFT is larger on 18 and 19 September (Table 2), when the UHII and UHIC are stronger and the BL wind speed is lower, than on other days (Table S2). On 14 September the BL wind speed was the highest and on 15 September the UHII was the weakest (Table S2), resulting in a substantially smaller UHI impact on the $PM_{2.5}$ concentrations on these days (Table 2). The UHIC effect consistently plays a dominant role (72% to 93%) in affecting the vertical distribution of $PM_{2.5}$ in all UHI cases (Table 2). During the 8-day experiment, the UHI effect increased the mean concentrations of $PM_{2.5}$, nitrate, ammonium, and sulfate in the LFT by 7%, 20%, 8%, and 3%, respectively (Table 2). This suggests that ammonium nitrate aerosols increased considerably in the LFT due to the UHI effects.

5) Please check the English and avoid the typo errors.

**Response:** Thank you for your suggestions. We double-checked the English spelling and made sure there were no typo errors.